# Urinary Tartaric Acid, a Biomarker of Wine Intake, Correlates with Lower Total and LDL Cholesterol

**DOI:** 10.3390/nu13082883

**Published:** 2021-08-22

**Authors:** Inés Domínguez-López, Isabella Parilli-Moser, Camila Arancibia-Riveros, Anna Tresserra-Rimbau, Miguel Angel Martínez-González, Carolina Ortega-Azorín, Jordi Salas-Salvadó, Olga Castañer, José Lapetra, Fernando Arós, Miquel Fiol, Lluis Serra-Majem, Xavier Pintó, Enrique Gómez-Gracia, Emilio Ros, Rosa M. Lamuela-Raventós, Ramon Estruch

**Affiliations:** 1Department of Nutrition, Food Science and Gastronomy, XIA, School of Pharmacy and Food Sciences, INSA, University of Barcelona, 08028 Barcelona, Spain; idominguez@ub.edu (I.D.-L.); iparillim@ub.edu (I.P.-M.); carancri77@alumnes.ub.edu (C.A.-R.); annatresserra@ub.edu (A.T.-R.); 2Centro de Investigación Biomédica en Red Fisiopatología de la Obesidad y la Nutrición (CIBEROBN), Instituto de Salud Carlos III, 28029 Madrid, Spain; mamartinez@unav.es (M.A.M.-G.); carolina.ortega@uv.es (C.O.-A.); jordi.salas@urv.cat (J.S.-S.); ocastaner@imim.es (O.C.); joselapetra543@gmail.com (J.L.); luisfernando.aros@ehu.eus (F.A.); miguel.fiol@ssib.es (M.F.); lserra@dcc.ulpgc.es (L.S.-M.); xpinto@bellvitgehospital.cat (X.P.); egomezgracia@uma.es (E.G.-G.); eros@clinic.cat (E.R.); 3Department of Preventive Medicine and Public Health, School of Medicine, University of Navarra, 31008 Pamplona, Spain; 4Department of Nutrition, Harvard TH Chan School of Public Health, Boston, MA 02115, USA; 5Department of Preventive Medicine and Public Health, School of Medicine, University of Valencia, 46010 Valencia, Spain; 6Unitat de Nutrició Humana, Departament de Bioquímica i Biotecnologia, Hospital Universitari San Joan de Reus, Universitat Rovira i Virgili, 43201 Reus, Spain; 7Institut d’Investigació Sanitària Pere Virgili (IISPV), 43201 Reus, Spain; 8Cardiovascular Epidemiology Unit, Municipal Institute for Medical Research (IMIM), 08007 Barcelona, Spain; 9Research Unit, Department of Family Medicine, Distrito Sanitario Atención Primaria Sevilla, 41010 Sevilla, Spain; 10Department of Cardiology, Hospital Txangorritxu, 01009 Vitoria, Spain; 11Institut Universitari d’Investigació en Ciències de la Salut (IUNICS), 07122 Palma de Mallorca, Spain; 12Department Clinical Sciences, University of Las Palmas de Gran Canaria, 35016 Palmas de Gran Canaria, Spain; 13Lipid Unit, Department of Internal Medicine, IDIBELL-Hospital Universitari de Bellvitge, L’Hospitalet de Llobregat, FIPEC, 08908 Barcelona, Spain; 14Department of Epidemiology, School of Medicine, University of Malaga, 29010 Málaga, Spain; 15Lipid Clinic, Endocrinology and Nutrition Service, Institut d’Investigacions Biomèdiques August Pi Sunyer (IDIBAPS), Hospital Clínic, 08036 Barcelona, Spain; 16Internal Medicine Department, Institut d’Investigacions Biomèdiques August Pi Sunyer (IDIBAPS), Hospital Clinic, University of Barcelona, 08036 Barcelona, Spain

**Keywords:** PREDIMED, Mediterranean diet, lipid profile, cardiovascular risk, polyphenols, menopause, body fat, biomarkers, tartaric acid

## Abstract

Postmenopausal women are at higher risk of developing cardiovascular diseases due to changes in lipid profile and body fat, among others. The aim of this study was to evaluate the association of urinary tartaric acid, a biomarker of wine consumption, with anthropometric (weight, waist circumference, body mass index (BMI), and waist-to-height ratio), blood pressure, and biochemical variables (blood glucose and lipid profile) that may be affected during the menopausal transition. This sub-study of the PREDIMED (Prevención con Dieta Mediterránea) trial included a sample of 230 women aged 60–80 years with high cardiovascular risk at baseline. Urine samples were diluted and filtered, and tartaric acid was analyzed by liquid chromatography coupled to electrospray ionization tandem mass spectrometry (LC-ESI-MS/MS). Correlations between tartaric acid and the study variables were adjusted for age, education level, smoking status, physical activity, BMI, cholesterol-lowering, antihypertensive, and insulin treatment, total energy intake, and consumption of fruits, vegetables, and raisins. A strong association was observed between wine consumption and urinary tartaric acid (0.01 μg/mg (95% confidence interval (CI): 0.01, 0.01), *p*-value < 0.001). Total and low-density lipoprotein (LDL) cholesterol were inversely correlated with urinary tartaric acid (−3.13 μg/mg (−5.54, −0.71), *p*-value = 0.016 and −3.03 μg/mg (−5.62, −0.42), *p*-value = 0.027, respectively), whereas other biochemical and anthropometric variables were unrelated. The results suggest that wine consumption may have a positive effect on cardiovascular health in postmenopausal women, underpinning its nutraceutical properties.

## 1. Introduction

Cardiovascular disease (CVD) is the leading cause of death worldwide in both sexes. Nevertheless, important sex-specific differences exist. According to the American Heart Association, menopause is listed as a female-specific cardiovascular risk factor (CVRF) [1]. During the menopause transition women experience adverse changes in their lipid profile, body fat distribution, metabolic syndrome risk, and vascular health [2,3,4,5]. Previous studies suggested that menopause is associated with increased total and low-density lipoprotein (LDL) cholesterol [6] and changes in body composition such as increased fat mass and loss of lean mass [7]. Changes in blood pressure (BP), waist circumference (WC), body mass index (BMI), and blood glucose and insulin have not been specifically associated with menopause and appear to reflect chronological aging [5,6,8]. Therefore, menopause-induced increases in cholesterol, body fat, and possibly other CVRFs may accelerate the risk of developing CVD.

Diet and lifestyle can also affect the incidence of CVD. Modifiable factors, such as smoking cessation, healthy diet, and regular physical activity, play a crucial role in reducing cardiovascular risk [9]. The Mediterranean diet (MedDiet) has been associated with a better control of several CVRFs [10] through improvements in BP, lipid profile, glucose metabolism, arrhythmia risk, and gut microbiome [11,12]. One of the main characteristics of the MedDiet is the abundant consumption of olive oil, vegetables, fruits, nuts, legumes, fish, and cereals, and moderate wine consumption [13,14]. Epidemiologic studies and randomized clinical trials reported that moderate consumption of wine (1 or 2 glasses/day) during meals has been consistently associated with a lower risk of CVD [15,16,17]. In the context of a MedDiet, moderate alcohol consumption appears to be synergistic with other cardioprotective components of the MedDiet that increase high-density lipoprotein (HDL) cholesterol, decrease platelet aggregation, promote antioxidant effects, and reduce inflammation [13].

Wine consumption is mainly determined through dietary questionnaires. A biomarker of wine intake reflects its consumption more reliably than a questionnaire, since people may not accurately report the amount of alcohol consumed due to perceived social rejection of excessive consumption [18]. Tartaric acid, the main organic acid in wine and the molecule responsible for wine acidity, is present in high amounts in wine (1.5–4.0 g/L) but is rare in most common foods [19,20]. Urinary tartaric acid has been considered as a sensitive, selective, and robust biomarker of moderate wine intake [21,22]. Therefore, determining tartaric acid stands out as a useful tool to further study the impact of moderate wine drinking on health. The aim of this study was to determine the association between urinary tartaric acid as a biomarker of wine consumption and CVRFs in postmenopausal women at risk of developing CVD.

## 2. Materials and Methods

### 2.1. Study Design

This study is a cross-sectional analysis of baseline data from a subsample of participants in the PREDIMED (PREvención con DIeta MEDiterránea) study, a large, parallel-group, multicenter, randomized, controlled, 5-year clinical trial conducted between 2003 and 2009. The objective was to assess the effect of a Mediterranean diet supplemented with extra-virgin olive oil or mixed nuts as compared to a low-fat diet on the primary prevention of CVD in 7447 participants at high cardiovascular risk. Eligible participants were men (55–80 years old) and women (60–80 years old) who had type 2 diabetes mellitus or at least 3 of the following major CVRFs: smoking, hypertension, elevated LDL cholesterol, low HDL cholesterol, overweight or obesity, and/or family history of premature coronary heart disease [23]. All participants provided written informed consent, and the study protocol and procedures complied the ethical standards of the Declaration of Helsinki.

For the present sub-study of the PREDIMED trial, urinary tartaric acid was analyzed in a subsample of women equivalent to 5% of the total female population of the PREDIMED study. The 230 women that were randomly selected had undergone the menopausal transition and their urine samples were available at baseline. Participants who had no available data of total energy intake or reported extreme values (>3500 kcal/day) were excluded from the analysis (*n* = 8).

### 2.2. Anthropometric, Dietary, and Physical Activity Assessments

Trained personal performed the anthropometric and clinical measurements (height, weight, WC, and BP). BMI was obtained by dividing the body weight in kilograms by the square of height in cm. The waist-to-height ratio (WtHR) was calculated by dividing the WC in centimeters by height in meters. Systolic (SBP) and diastolic blood pressure (DBP) were measured in triplicate with a validated semi-automatic oscillometer (Omron HEM-705CP, Lake Forest, IL, USA). A validated semi-quantitative food frequency questionnaire (FFQ), which included 137 food items [24], and the Minnesota Leisure-Time Physical Activity Questionnaire [25] were used to assess dietary habits over the previous 12 months and physical activity (metabolic equivalent tasks per minute per day, METs min/day) of the participants.

### 2.3. Clinical Measurements

Medical conditions, family history of disease, and risk factors were collected though a questionnaire during the first screening visit. Biological samples (plasma and urine) were collected at baseline after 12 h overnight fast and stored at *−*80 °C until assay. Blood glucose, total cholesterol, triglycerides, and HDL cholesterol were determined by standard enzymatic methods, and LDL cholesterol was calculated by the Friedewald equation [26].

### 2.4. Tartaric Acid Determination

#### 2.4.1. Reagents and Standards

Formic acid (approximately 98%), picric acid (98%, moistened with approximately 33% water), and sodium hydroxide (≥98%) were obtained from Panreac. L-(+)-Tartaric acid and creatinine were purchased from Sigma. The labelled internal standard DL-(±)-tartaric-2,3-d2 acid was obtained from C/D/N Isotopes. Solvents were high-performance liquid chromatography grade, and all other chemicals were analytical reagent grade. Ultrapure water was obtained from a Milli-Q Gradient water purification system (Millipore, Bedford, MA, USA).

Stock solutions of tartaric acid were prepared in water. Working standard solutions that ranged from 0.01 to 5 µg/mL were made by appropriate dilution in 0.5% formic acid in water and then stored in amber glass vials at −20 °C.

#### 2.4.2. Sample Preparation

Determination of urinary tartaric acid was performed following a previously validated stable-isotope dilution LC-ESI-MS/MS method by our research group [27]. Briefly, urine samples (20 µL) were diluted 1:50 (*v*:*v*) with 0.5% formic acid in water, and 10 µL of a deuterated isotope standard solution in water (DL-(±)-tartaric-2,3-d2 acid, 200 µg/mL) were added. The sample dilution was passed through a 0.20 µm filter and analyzed by LC–ESI-MS/MS. Urinary tartaric acid data were corrected by urine creatinine, measured according to the adapted Jaffé alkaline picrate method for thermo microtiter 96-well plates, according to Medina-Remón et al. [28]. Finally, urinary tartaric acid was expressed as µg of tartaric acid per mg of creatinine. According to previous data, the cut-off of 8.84 μg/mg creatinine was used to discriminate daily consumers and non-consumers of wine [21].

#### 2.4.3. LC–ESI-MS/MS Analysis

After filtration, tartaric acid was analyzed using an Atlantis TE C18, 100 mm × 2.1 mm, 3 µm (Waters, Milford, MA, USA) reversed-phase column coupled for detection to the triple quadrupole mass spectrometer API 3000 (Applied Biosystems, Foster City, CA, USA). The mass spectrometer was operated in negative electrospray ionization mode. The column was maintained at 25 °C throughout the analysis. Mobile phases A and B were 0.5% formic acid in water and 0.5% formic acid in acetonitrile, respectively. The following linear gradient was used: holding at 100%A for 3.5 min, decrease to 10%A over 2 min and holding for 2 min, return to initial conditions for 1.5 min, and re-equilibration for 6 min. The flow rate was set at 350 µL/min and the injection volume was 10 µL. Post-column addition of acetonitrile (250 µL/min) was carried out to improve analyte ionization efficiency. The detection was accomplished in multiple reaction monitoring (MRM) mode, and the following MS/MS transitions were used for quantification and confirmation, respectively: m/z 149/87 and m/z 149/73 for tartaric acid, and m/z 151/88 and m/z 151/74 for the deuterated isotope.

### 2.5. Statistical Analyses

The baseline characteristics of participants are presented as means and standard deviations (SD) for continuous variables, and frequency (*n*) and percentage (%) for categorical variables.

The normality of continuous variables was assessed with the Shapiro–Wilk test. The variables without normal distribution were transformed into logarithms. Multiple adjusted linear regression models were used to assess the differences between urinary tartaric acid and wine consumption as well as anthropometric and biochemical measurements. Three different adjustment models were applied. Model 1 was minimally adjusted for age (continuous). Model 2 was additionally adjusted for educational level, smoking status, BMI (except for anthropometric criteria), physical activity, and cholesterol-lowering, antihypertensive, and insulin treatment. Model 3 was further adjusted for total energy intake and consumption of fruits, vegetables, and raisins. We used robust variance estimators to account for the recruitment center in all linear models. To illustrate the relationship between wine consumption and urinary tartaric acid, the mL per month of wine reported in the FFQ were transformed into glasses of wine (with 1 glass equivalent to 100 mL).

Values are shown as 95% confidence interval (CI) and significance for all statistical tests was based on bilateral contrast set at *p* < 0.05. All the statistical analyses were performed using Stata statistical software package version 16.0 (StataCorp LP, College Station, TX, USA).

## 3. Results

### 3.1. Study Population

The main characteristics of the PREDIMED participants who were included in this sub study are summarized in Table 1. The mean age of the women was 66.9 + 0.4 years. Their burden of CVRFs was high: 42.1% had been diagnosed with type 2 diabetes, 87.3% with hypertension, and 76.5% with hypercholesterolemia. Among the drug treatments, statins were the most common medication, with 40.72% of them under treatment. Furthermore, 9.1% of the participants were current smokers. Finally, 82% of the participants had a low educational level.

Up to 45.7% of the participants reported wine consumption in the FFQ. The mean concentration of tartaric acid in urine was 28.34 μg/mg creatinine, and 40.4% were considered daily consumers of wine.

The mean values of anthropometric measurements revealed that most participants were obese, as defined by their BMI, WC, and WtHR data [29] according to the International Diabetes Federation and the American Heart Association [30]. Regarding biochemical measurements, triglycerides and HDL cholesterol were at desirables levels, while total cholesterol, LDL cholesterol and glucose were borderline high [31,32].

The mean energy intake was 2161 kcal/day, of which carbohydrates accounted for 42.0% of the energy consumed, protein intake 16.8%, and fat intake 39.8%.

### 3.2. Tartaric Acid as a Biomarker of Wine Consumption

After adjustments for several covariates (age, education level, smoking status, physical activity, BMI, cholesterol-lowering, antihypertensive, and insulin treatment, total energy intake, and consumption of fruits, vegetables, and raisins), women who consumed more wine presented higher concentrations of tartaric acid in urine (0.01 μg/mg (95% CI: 0.01, 0.01), *p*-value < 0.001). Figure 1 illustrates the linear relationship between urinary tartaric acid concentrations and wine consumption expressed as glasses of wine, excluding those who reported not consuming wine.

### 3.3. Anthropometric Measurements and Urinary Tartaric Acid

After adjustment for several covariates, we observed no association between urinary tartaric acid and BMI, WC, weight, WtHR, and systolic or diastolic BP (Table 2).

### 3.4. Biochemical and Clinical Measurements and Urinary Tartaric Acid

A negative association was observed between urine tartaric acid and total and LDL cholesterol after full adjustment (−3.13 μg/mg (−5.54, −0.71), *p*-value = 0.016 and −3.03 μg/mg (−5.62, −0.42), *p*-value = 0.027, respectively). By contrast, no differences were observed for HDL with different concentrations of tartaric acid. Finally, no association was found between triglycerides and glucose and tartaric acid concentrations (Table 3).

## 4. Discussion

In this sub-analysis of a subset of postmenopausal women participating in the PREDIMED trial, urinary tartaric acid concentrations as an objective biomarker of wine intake were significantly associated with lower concentrations of total and LDL cholesterol. No associations with anthropometric variables or blood pressure were observed. To the best of our knowledge, the current study is the first to evaluate wine consumption based on a biomarker in a postmenopausal population at increased risk of developing CVD.

Wine consumption has been widely studied due to its beneficial effects on cardiovascular and metabolic health [10]. However, most studies have evaluated wine intake using FFQs or self-questionnaires instead of biological biomarkers, a more reliable and objective way of assessing dietary habits [33]. It has been previously demonstrated that urinary tartaric acid is a specific and sensitive biomarker, as its major sources in the diet are grapes and wine [21,34]. Accordingly, we observed a positive association between wine consumption reported in the FFQs and the concentrations of tartaric acid present in urine. Other phenolic compounds, such as resveratrol and its metabolites, have been proposed as wine biomarkers. However, the resveratrol content in wine is subject to a high variability and its metabolism shows interindividual differences [35]. Thus, selectivity and high correlation with reported intakes make tartaric acid a reliable dietary biomarker of wine consumption.

Different studies have evaluated how alcohol intake affects different parameters of body composition. A cross-sectional study in French adults suggested an inverse association in women of wine intake 100 g/day with BMI and WtHR [36]. Tresserra-Rimbau et al. analyzed the effects of red wine consumption on the prevalence of metabolic syndrome and its components, and found a negative association between moderate red wine consumption and BMI [16]. Tolstrup et al. also described inverse associations between alcohol consumption and WC in women [37], while other studies found no relationship between alcohol consumption and body weight in women [38,39]. The mentioned literature indicates that moderate consumption of wine, an alcoholic beverage that contributes to energy intake, is not related to weight gain or detrimental changes in body composition. Our study supports this notion, as we did not observe any differences in BMI, weight, WC, and WtHR with increasing wine consumption.

Evaluating the effect of alcohol, and specifically wine, on the risk of developing CVD in women is important due to the increase in cardiovascular risk after menopause. Among CVRFs, a recent metanalysis reported that triglycerides, total cholesterol, LDL cholesterol, and the total cholesterol-to-HDL-cholesterol ratio were significantly higher in postmenopausal women compared to premenopausal women, and suggested that age was partly responsible for the differences in lipid levels [40]. We found that women with higher concentrations of tartaric acid presented lower total and LDL cholesterol. Similarly to our results, Rifler et al. reported that after 2 weeks of drinking 250 mL of red wine daily, patients post myocardial infarction presented a 5% decrease in total and LDL cholesterol [41]. Furthermore, Taborsky et al. evaluated the effect of 1 year of wine consumption, and observed a reduction in total and LDL cholesterol [42]. In another clinical trial, authors reported a similar beneficial effect on the lipid profile after consumption of red wine in asymptomatic hypercholesterolemic individuals [43]. The data are almost consistent in showing that wine consumption reduces LDL cholesterol while increasing HDL cholesterol [44,45]. Moderate consumption of alcohol has been associated with higher concentrations of HDL cholesterol and diminished lipid oxidation stress [46]. Resveratrol metabolites in urine, as biomarkers of wine consumption, were significantly associated with lower triglycerides and higher HDL-cholesterol [47]. However, we were unable to confirm that higher urinary tartaric acid as a biomarker of wine consumption was associated with raised HDL cholesterol levels. A probable reason is that the women studied had rather high baseline HDL cholesterol levels, making it more difficult to further increase these with interventions.

Many clinical studies support that light to moderate alcohol consumption, in particular of red wine, is associated with lower CVD rates and an improved lipid profile and inflammatory system [17,48]. However, it remains unknown whether this effect of wine is due to alcohol per se, the phytochemicals of wine, their combined effect, or even the time of drinking, since postprandial oxidative stress after a meal appears to be counteracted by the ingestion of red wine [49]. In this sense, it has been found that wine micro-constituents modulate inflammatory mediators and therefore may be responsible for attenuating postprandial inflammation [50]. In addition, they protect against the effect of ethanol on cytokine secretion, which are involved in inflammatory processes [51]. In support of this view, a randomized clinical trial reported that wine bioactive compounds, such as resveratrol, can decrease total cholesterol by reducing mRNA expression of hepatic 3-hydroxy-3-methyl-glutaryl-CoA (HMG-CoA) reductase, in addition to the increased activation of the sirtuin system in all tissues [52]. Experimental work in cell cultures and animal models has shown that the enhancement of Sirtuin 1 can lead to better metabolic profiles and anti-inflammatory activities, as well as increased reverse cholesterol transport [53]. Overall, evidence supports that wine micro-constituents play a crucial role in the protective effect of wine on cardiovascular health by exerting anti-inflammatory actions.

The main strength of this study is that it used a biological biomarker, tartaric acid, to evaluate wine consumption, instead of less reliable methods such as FFQs or self-reported questionnaires. Moreover, it involved baseline data of participants in the PREDIMED trial; therefore, the results reflect real-life conditions. The main limitations were the modest sample size and the impossibility of determining causality due to the cross-sectional design.

## 5. Conclusions

The findings from the current cross-sectional study support the notion that wine intake has beneficial nutraceutical effects on the cardiovascular health of postmenopausal women, as its biomarker tartaric acid was associated with lower total and LDL cholesterol concentrations. Randomized trials are needed to confirm these results and determine the impact of wine consumption on cardiovascular health in a sensitive population such as that of postmenopausal women.

## Figures and Tables

**Figure 1 nutrients-13-02883-f001:**
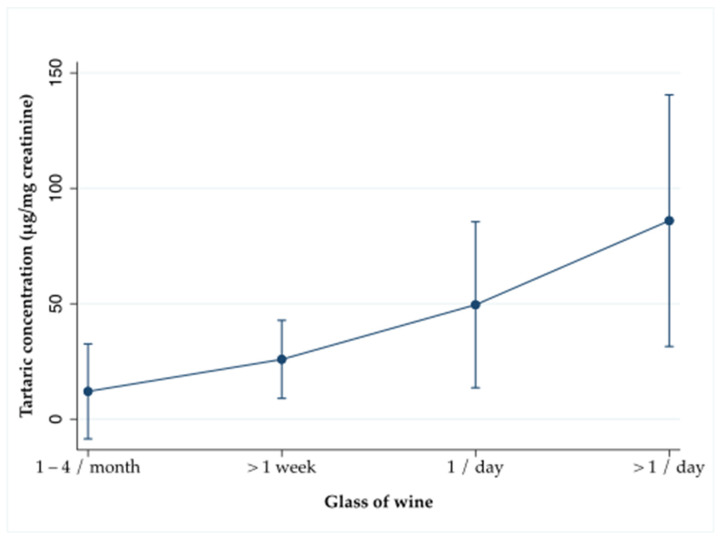
Relationship between urinary tartaric acid concentrations in urine and wine consumption expressed as glasses of wine.

**Table 1 nutrients-13-02883-t001:** Baseline characteristics of the women in the study population (*n* = 222).

General Characteristics	
Age, years	66.9 ± 0.4
Type 2 diabetes, *n* (%)	93 (42.08)
Hypertension, *n* (%)	193 (87.34)
Hypercholesterolemia, *n* (%)	169 (76.55)
Medication use, *n* (%)	
ACE inhibitors	64 (28.96)
Diuretics	58 (26.24)
Statins	90 (40.72)
Other lipid-lowering agents	14 (6.33)
Insulin	10 (4.52)
Oral hypoglycemic agents	53 (23.98)
Antiplatelet therapy	46 (20.81)
Current smoker, *n* (%)	20 (9.05)
Leisure-time physical activity, MET·min/week	186.5 ± 10.9
Educational level, *n* (%)	
Low	182 (82.35)
Medium	24 (10.86)
High	15 (6.79)
Wine consumption, *n* (%)	101 (45.70)
Urinary tartaric acid, μg/mg creatinine	28.47 + 4.03
Daily wine consumers, *n* (%)	81 (40.4)
**Anthropometric measurements, mean + SD**	
Weight, kg	72.9 ± 0.7
BMI, kg/m^2^	30.3 ± 0.28
WC, cm	97.5 ± 0.7
WtHR	63.0 ± 0.5
**Biochemical measurements, mean + SD**	
Total cholesterol, mg/dL	221.0 ± 2.7
LDL cholesterol, mg/dL	136.5 ± 2.4
HDL cholesterol, mg/dL	57.7 ± 1.0
Triglycerides, mg/dL	134.0 ± 5.2
Glucose, mg/dL	118.1 ± 2.4
**Blood pressure, mean + SD**	
Systolic, mm Hg	148.7 ± 1.2
Diastolic, mm Hg	83.8 ± 0.6
**Dietary intake, mean + SD**	
Total energy, kcal/day	2161 ± 33
Carbohydrate, % of energy	42.0 ± 0.5
Protein, % of energy	16.8 ± 0.2
Fat, % of energy	39.8 ± 0.5

ACE: angiotensin-converting enzyme; MET: metabolic equivalent task; BMI: body mass index; WC: waist circumference; WtHR: waist-to-height ratio; LDL: low-density lipoprotein cholesterol; HDL: high-density lipoprotein cholesterol. Data are expressed as the mean ± standard deviations (SD) for continuous variables and frequency (*n*) and percentage (%) for categorical variables.

**Table 2 nutrients-13-02883-t002:** Association between anthropometric variables and urinary tartaric acid (μg/mg creatinine).

		β (95% IC)	*p-*Value
BMI, kg/m^2^	Model 1	<−0.01 (−0.01, 0.01)	0.519
	Model 2	<−0.01 (−0.01, 0.01)	0.426
	Model 3	<−0.01 (−0.01, 0.01)	0.973
WC, cm	Model 1	0.56 (−0.25, 1.38)	0.173
	Model 2	0.61 (−0.04, 1.26)	0.064
	Model 3	0.70 (−0.12, 1.51)	0.087
Weight, kg	Model 1	<−0.01 (−0.01, 0.01)	0.770
	Model 2	<−0.01 (−0.01, 0.01)	0.766
	Model 3	<0.01 (−0.01, 0.01)	0.886
WtHR	Model 1	0.29 (−0.25, 0.82)	0.291
	Model 2	0.32 (−0.17, 0.81)	0.175
	Model 3	0.41 (−0.14, 0.96)	0.128

BMI: body mass index; WC: waist circumference; WtHR: waist-to-height ratio; CI: confidence interval. Regression coefficients (95%CI) were obtained from multivariable adjusted linear regression models. β: Non-standardized coefficient. Model 1: adjusted for age. Model 2: adjusted for age, educational level, smoking status, physical activity, and cholesterol-lowering, antihypertensive, and insulin treatment. Model 3: adjusted for age, educational level, smoking status, physical activity, cholesterol-lowering, antihypertensive, and insulin treatment, total energy intake, and consumption of fruits, vegetables, and raisins. We used robust standard errors to account for recruitment center. *p*-values < 0.05 were considered significant.

**Table 3 nutrients-13-02883-t003:** Association between biochemical variables and blood pressure and urinary tartaric acid (μg/mg creatinine).

		β (95% CI)	*p-*Value
Total cholesterol, mg/dL	Model 1	−3.32 (−6.53, −0.10)	0.043
	Model 2	−3.24 (−5.78, −0.72)	0.017
	Model 3	−3.13 (−5.54, −0.71)	0.016
LDL cholesterol, mg/dL	Model 1	−3.44 (−6.34, −0.53)	0.021
	Model 2	−3.43 (−5.86, −1.00)	0.010
	Model 3	−3.03 (−5.62, −0.42)	0.027
HDL cholesterol, mg/dL	Model 1	<−0.01 (−0.02, 0.02)	0.689
	Model 2	−0.01 (−0.03, 0.01)	0.220
	Model 3	<−0.01 (−0.02, 0.01)	0.633
Triglycerides, mg/dL	Model 1	0.01 (−0.04, 0.05)	0.739
	Model 2	0.02 (−0.03, 0.07)	0.422
	Model 3	0.02 (−0.04, 0.07)	0.525
Glucose, mg/dL	Model 1	0.02 (<−0.01, 0.04)	0.092
	Model 2	0.03 (−0.01, 0.07)	0.119
	Model 3	0.02 (−0.01, 0.06)	0.180
Systolic BP, mmHg	Model 1	−0.34 (−1.79, 1.11)	0.647
	Model 2	−0.09 (−1.53, 1.71)	0.904
	Model 3	0.36 (−1.27, 1.99)	0.633
Diastolic BP, mmHg	Model 1	0.05 (−0.71, 0.81)	0.893
	Model 2	0.24 (−0.47, 0.95)	0.472
	Model 3	0.21 (−0.47, 0.89)	0.502

BP: blood pressure; LDL: low-density lipoprotein cholesterol; HDL: high-density lipoprotein cholesterol; CI: confidence interval. Regression coefficients (95%CI) were obtained from multivariable adjusted linear regression models. β: Non-standardized coefficient. Model 1: adjusted for age. Model 2: adjusted for age, educational level, smoking status, physical activity, BMI, and cholesterol-lowering, antihypertensive, and insulin treatment. Model 3: adjusted for age, educational level, smoking status, physical activity, cholesterol-lowering, antihypertensive, and insulin treatment, total energy intake, and consumption of fruits, vegetables, and raisins. We used robust standard errors to account for recruitment center. *p*-values < 0.05 were considered significant.

## Data Availability

There are restrictions on the availability of data for the PREDIMED trial due to the signed consent agreements around data sharing, which only allow access to external researchers for studies following project purposes. Requestors wishing to access the PREDIMED-Plus trial data used in this study can make a request to the PREDIMED trial Steering Committee chair: restruch@clinic.cat. The request will then be passed to members of the PREDIMED Steering Committee for deliberation.

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
