# Peer review of "Urinary Tartaric Acid, a Biomarker of Wine Intake, Correlates with Lower Total and LDL Cholesterol"

_nutrients, 2021, doi:10.3390/nu13082883_

Round 1
Reviewer 1 Report
In this paper “Urinary tartaric acid, a biomarker of wine intake, correlates with lower total and LDL cholesterol”. Authors have evaluated the association of urinary tartaric acid, a biomarker of wine consumption, with anthropometric, blood pressure, and biochemical variables. A strong association was observed between wine consumption and urinary tartaric acid, total and LDL-cholesterol were inversely correlated with urinary tartaric acid whereas other biochemical and anthropometric variables were unrelated.
This work could be publishable after major correction. The data presentation needs to be improved. I have few comments and recommend the authors to address. My comments are given below:
- Include separate section for biological samples and enzymatic assays in the materials and method sections. To keep it separate from physical measurements. Lines 124-127.
- Lines 137-138, include concentrations of the standards instead of different working standards.
- Check paper thoroughly for English editing. For instance, lines 154-155 the word respectively should come in the end of sentence.
- Positive or negative ionization mode was used? Mention in the respective method section.
- What is PREDIMED study? Is there any substitution for this word, which can be well understood easily, without any reference?
- Some sentences are long e.g., lines 98-101. Write down small and concise sentences.
- Provide a flow chart of study design to make a clear indication of the groups, sex, number of subjects etc.
- Illustrate the total cholesterol, triglycerides, HDL-cholesterol, glucose results in bar graphs and provide their correlations with tartaric acid as a graphical representation.
- What is the typical concentration of tartaric acid in the wine? What is use of tartaric acid in the wine?
Reviewer 2 Report
The aim of this study was to assess the was to determine the association between urinary tartaric acid as a biomarker of wine consumption and several risk factors for CVD in postmenopausal women. It is a very interest topic and I think that the food biomarkers are the future in the assignment of dietary habits since if the analytical methods is well validated it would be more accurate than traditional dietary habits recording methods. However, I believe that some data must be added.
- The authors should refer in the study design the chronological period that urine collection was performed, since the time of the urine storage is important for biomarkers’ stability.
- Please check the SD of age. Is the range of the age of the volunteers so narrow?
- Were the volunteers under medication except cholesterol-lowering treatment? The authors should refer more details about the medication in the study design section.
- My suggestion is to refer the % of carbohydrate, protein and fat in Table 1.
- The authors should include in the method section the intra and inter assay variation of tartaric measurement. Also, the detection limit of their method and the range of the values of the tartaric standard curve.
- The authors refer that “…excluding those who reported not consuming wine” for the data presenting in fig 1. I was wondering if the values of tartaric in these volunteers were under detection limit. Was there any false positive value? Did the values of tartaric also correlate with fruit consumption (tartaric also exist in grapes)? I think that the authors should clarify these issues and provide the values of tartaric also in those that reported not consuming wine
- My main concern is that the authors present data concerning only the effect of wine consumption on the lipid profile and glucose of the volunteers. First of all, I think insulin levels and an index of insulin resistant and their correlations with tartaric levels should be added (especially since the 93% of volunteers are diabetic). Secondly, although lipid profile is the most establish risk factor for CVDs it is the simplest approach concerning the mechanisms of beneficial effects of moderate wine consumption. It is known that wine micro-constituents exert potent in vitro anti-inflammatory effects, as the authors refer in the discussion section, and some clinical trials also revealed the same conclusion (add ref: Metabolism, 2018;83:102-119 doi: 10.1016/j.metabol.2018.01.024). Therefore, I believe that the authors should add some data concerning inflammatory mediators (that already been measured in PREDIMED study) and present the impact of wine consumption (via tartaric levels) in inflammation. These results I think will improve the outputs of the study and will go further than the classic risk factors (lipid profile, insulin resistance, and body composition).
- The authors also state that “However, it remains unknown whether this effect of wine is due to alcohol per se, the phytochemicals of wine, their combined effect, or even the time of drinking, since postprandial oxidative stress after a meal appears to be counteracted by the ingestion of red wine”. They should also consider in their discussion that there are studies that provide evidence that wine micro-constituents, but not the ethanol, exert postprandial (Prostaglandins Other Lipid Mediat. 2017 May;130:23-29. doi: 10.1016/j.prostaglandins.2017.03.002) as well as long-term (Cytokine 2021 Jul 8;146:155629.doi: 10.1016/j.cyto.2021.155629.) anti-inflammatory actions. In addition, in some cases seems to counteract the harmful effects of ethanol.
Round 2
Reviewer 1 Report
Authors have responded well to my comments.